# The Use of Spectroscopic Methods to Study Organic Matter in Virgin and Arable Soils: A Scoping Review

Evgeny Lodygin [1,*] and Evgeny Abakumov [2]

1 Institute of Biology, Komi Science Center, Ural Branch, Russian Academy of Sciences, 167982 Syktyvkar, Russia
2 Department of Applied Ecology, Faculty of Biology, Saint Petersburg State University, 7/9 Universitetskaya nab., 199034 St. Petersburg, Russia; e_abakumov@mail.ru or e.abakumov@spbu.ru
* Correspondence: lodigin@ib.komisc.ru

**Abstract:** The use of modern spectroscopic methods of analysis, which provide extensive information on the chemical nature of substances, significantly expands our understanding of the molecular composition and properties of soil organic matter (SOM) and its transformation and stabilization processes in various ecosystems and geochemical conditions. The aim of this review is to identify and analyze studies related to the application of nuclear magnetic resonance (NMR) and electron paramagnetic resonance (EPR) spectroscopy techniques to study the molecular composition and transformation of organic matter in virgin and arable soils. This article is mainly based on three research questions: (1) Which NMR spectroscopy techniques are used to study SOM, and what are their disadvantages and advantages? (2) How is the NMR spectroscopy technique used to study the molecular structure of different pools of SOM? (3) How is ESR spectroscopy used in SOM chemistry, and what are its advantages and limitations? Relevant studies published between 1996 and 2024 were searched in four databases: eLIBRARY, MDPI, ScienceDirect and Springer. We excluded non-English-language articles, review articles, non-peer-reviewed articles and other non-article publications, as well as publications that were not available according to the search protocols. Exclusion criteria for articles were studies that used NMR and EPR techniques to study non-SOM and where these techniques were not the primary methods. Our scoping review found that both solid-state and solution-state NMR spectroscopy are commonly used to study the structure of soil organic matter (SOM). Solution-phase NMR is particularly useful for studying soluble SOM components of a low molecular weight, whereas solid-phase NMR offers advantages such as higher $^{13}C$ atom concentration for stronger signals and faster analysis time. However, solution-phase NMR has limitations including sample insolubility, potential signal aggregation and reduced sensitivity and resolution. Solid-state NMR is better at detecting non-protonated carbon atoms and identifying heterogeneous regions within structures. EPR spectroscopy, on the other hand, offers significant advantages in experimental biochemistry due to its high sensitivity and ability to provide detailed information about substances containing free radicals (FRs), aiding in the assessment of their reactivity and transformations. Understanding the FR structure in biopolymers can help to study the formation and transformation of SOM. The integration of two- and three-dimensional NMR spectroscopy with other analytical methods, such as chromatography, mass spectrometry, etc., provides a more comprehensive approach to deciphering the complex composition of SOM than one-dimensional techniques alone.

**Keywords:** humic substances; organic fertilizers; NMR spectroscopy; EPR spectroscopy

## 1. Introduction

The use of modern instrumental high-precision physicochemical methods of analysis, which provide extensive information on the chemical nature of organic matter, greatly enhances our understanding of the composition and properties of soil organic matter (SOM) and its transformation and stabilization processes in various ecosystems and geochemical conditions [1–4]. Many of the spectroscopic methods previously used in the visible, UV

and infrared (IR) range of electromagnetic waves provide indirect or difficult-to-decipher information on the molecular structure of those complex natural compounds that form the basis of SOM [5,6]. Methods such as nuclear magnetic resonance (NMR) and electron paramagnetic resonance (EPR) spectroscopy provide information on both the qualitative set of major atomic groups and the specific configuration of individual molecular fragments and functional groups. These physicochemical methods make it possible to study not only dissolved substances or solid preparations but also compounds that have an intermediate state (gels, colloids, etc.) [7–10].

The NMR method for studying the basic element of organic chemistry, carbon, began to be widely used in physics and chemistry in the 1960s [7]. Since the mid-1960s, NMR spectroscopy has become one of the most frequently used methods for studying the molecular structure of organic substances, primarily due to its high informative capability and relative ease of obtaining results [11].

Initially, the NMR method was developed for the study of hydrogen nuclei, and its use for the study of SOM was inhibited by the need for deuterated solvents and the difficulty of accounting for the processes of isotopic exchange between the hydrogen of humic substances (HSs) and the deuterium of the solution. However, as early as 1963, a short article [12] appeared that described the results of a $^1$H NMR study of a preparation of methylated HSs. A detailed review of $^1$H NMR studies can be found in a number of papers [13,14]. It should be noted that in the $^1$H NMR spectra, only the region of protons in aliphatic structures is well resolved. In the region of protons of aromatic structures, a very broad signal is usually observed, which is very difficult to interpret. This phenomenon is apparently caused by the high polydispersity of the HS molecules [15,16].

The natural content of $^{13}$C nuclei is extremely low, less than 1%. Therefore, their observation in the solid phase is difficult, but the problem has been solved with the development of modern emulsion NMR methods. In the late 1980s, there was a qualitative leap in the application of NMR to biochemical studies. The CP/MAS method for recording solid-phase NMR spectra was developed. It was based on polarization transfer from $^1$H nuclei to $^{13}$C (CP—cross polarization) and rotation of the sample at a "magic" angle (MAS—magic angle spinning), which made it possible to obtain satisfactory NMR spectra of solid-phase preparations in terms of sensitivity and resolution [17].

After E.K. Zavoysky discovered the phenomenon of electron paramagnetic resonance in 1944 [18], EPR spectroscopy provided information on the molecular structure of many inorganic and organic substances. Already, the first publication concerning the use of the EPR method in the study of biological systems showed that free radicals (FRs) can be contained in various biological objects, and a certain correlation between the metabolic activity of tissues and the concentration of FRs was established [19]. It turned out to be much more difficult to identify which specific FRs were involved in the processes under study.

The main advantage of the above methods is the ability to study not only SOM preparations but also soils in general, practically without affecting the object in any way. This makes it possible to solve the problem of correspondence between native substances in the composition of soils and isolated preparations of SOM [20,21]. Compared to other methods, NMR and EPR spectroscopy provide direct information about the structure of the compounds under investigation and are probably two of the most powerful methods for the comprehensive characterization of the structure of complex organic substances.

An analysis of the literature on the use of NMR and EPR spectroscopy in the study of the molecular composition of different SOM pools shows that researchers use different techniques and conditions for the acquisition of spectra. At the same time, they very often do not use the full potential of these methods, e.g., EPR spectroscopy is only used to estimate the total FR content. Also, the authors do not always take into account the limitations of the methods. All this has contributed to the writing of this scoping review to help researchers to choose the right conditions and techniques of NMR and EPR spectroscopy and to avoid mistakes in the interpretation of the results obtained.

Our aim was to identify and analyze studies related to the application of the NMR and EPR spectroscopy techniques to study the molecular composition and transformation of organic matter in virgin and arable soils. This review article is mainly based on three research questions: (1) Which NMR spectroscopy techniques are used to study SOM, and what are their disadvantages and advantages? (2) How is the NMR spectroscopy technique used to study the molecular structure of different pools of SOM? (3) How is ESR spectroscopy used in SOM chemistry, and what are its advantages and limitations?

## 2. Methods

The protocol was drafted and the review was reported following the PRISMA-ScR (preferred reporting items for systematic reviews and meta-analyses extension for scoping reviews) [22]. We chose the scoping review protocol to scope the literature on our topic and provide a comprehensive overview of the available literature and its focus.

Relevant studies were searched in four databases: eLIBRARY, MDPI, ScienceDirect and Springer (Figure 1). Keywords were searched in the title of articles, abstract and keywords. The following keywords were used: "organic matter" and "humic" in combination with "soil", "nuclear magnetic resonance", "electron paramagnetic resonance", "electron spin resonance" and their abbreviations "NMR", "EPR" and "ESR". Moreover, we considered only papers published from 1 January 1996 to 1 March 2024.

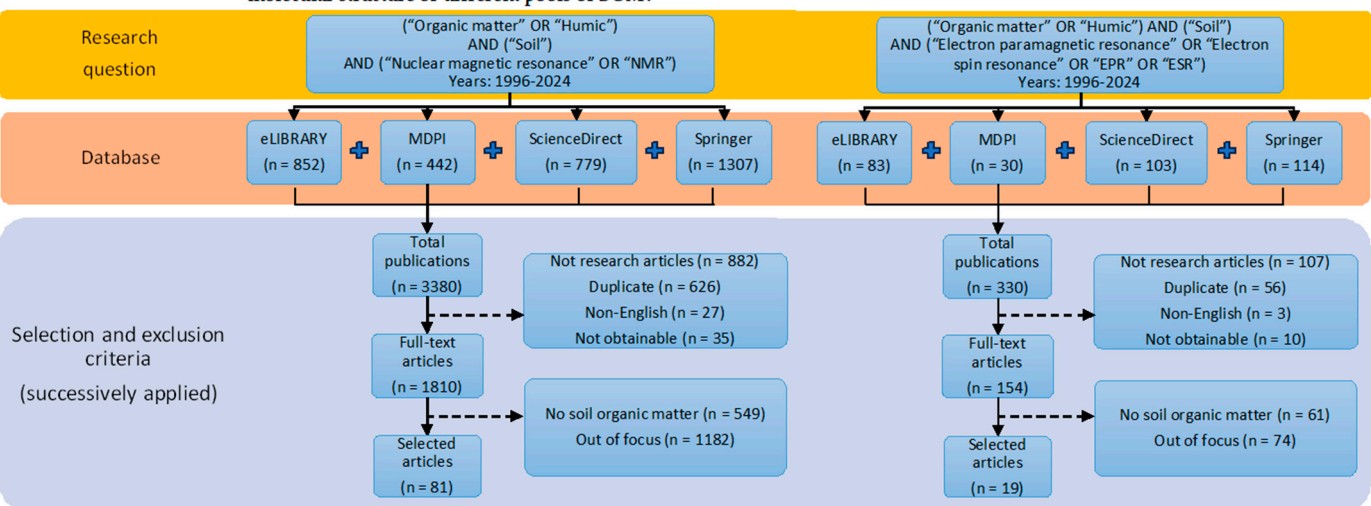

**Figure 1.** Flowchart of the search strategy and eligibility criteria.

The results of the literature search were exported to Endnote, which was used to detect and remove duplicates. We excluded "non-English"-language and "non-peer-reviewed" articles and other "non-article publications" or publications that were "not obtainable" by the search protocols. Review articles were also excluded from the analysis but were used for reference screening.

Exclusion criteria for articles were studies in which NMR and EPR spectroscopy techniques were used to study "non-soil organic matter" and where these techniques were not the primary ("out of focus") methods. Finally, we identified 100 relevant articles (Figure 2) and selected them based on the PRISMA-ScR guidelines.

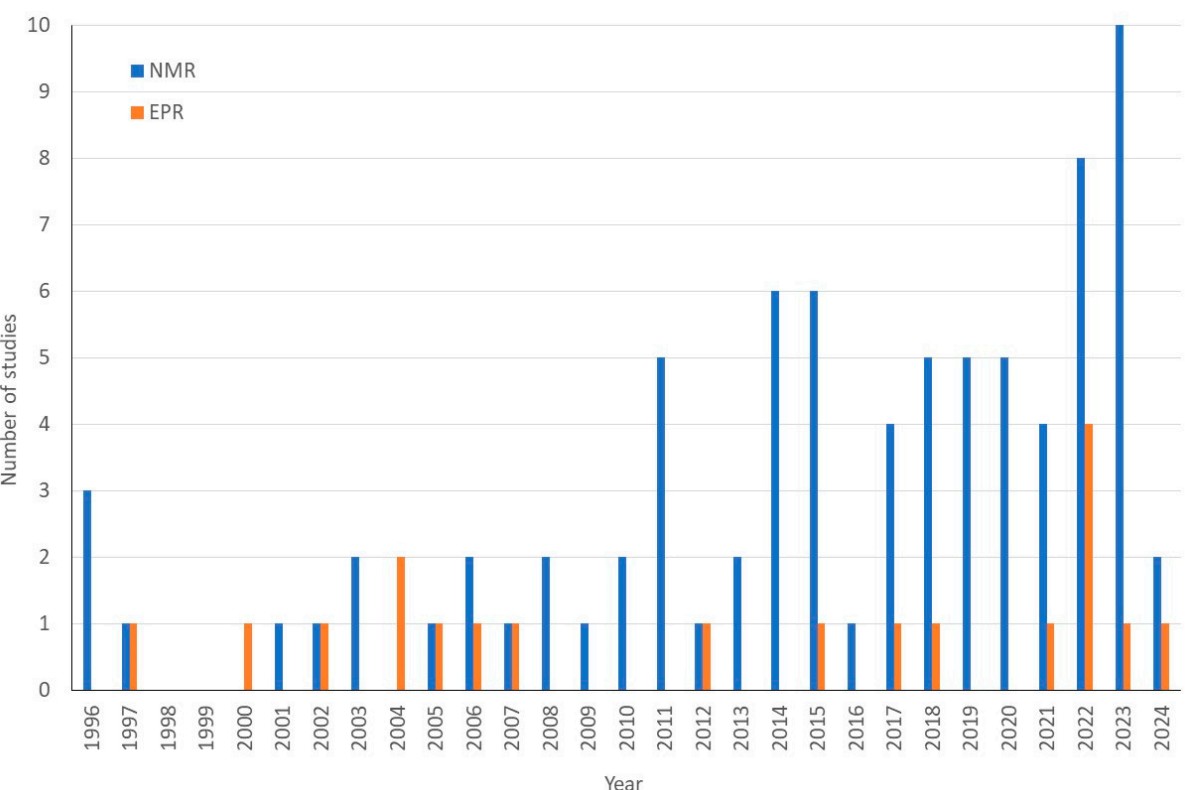

**Figure 2.** Annual distribution of the number of included articles on the use of NMR and EPR spectroscopy techniques to study the molecular structure of SOM.

The research questions posed in the review (Figure 1) were categorized to describe the qualitative information gathered from the articles:

(1)    Question—(i) the atoms studied within the SOM and (ii) the techniques (phases and multidimensional) used in the NMR;
(2)    Question—(i) interpretation of the NMR spectra and (ii) the SOM pools studied;
(3)    Question—(i) nature of SOM paramagnetism and (ii) the SOM pools of virgin and arable soils and parameters of the EPR spectra.

The categories were complementary (together, they explained the totality of the research). Each category had subcategories with varying degrees of specificity. An article may have different information for the same category or subcategory if it studied two or more pools of SOM or used multiple NMR and EPR techniques.

## 3. NMR Spectroscopy

The search for publications on the use of NMR spectroscopy to study SOM yielded 3380 records. After removing duplicates and excluding non-article, non-English and non-peer-reviewed publications, 1810 records remained suitable for further screening (Figure 1). The screening of article abstracts resulted in 81 records being retained for further analysis according to the pre-defined inclusion criteria. Most of the excluded full-text articles did not meet the scope of the review. This was because they did not specifically focus on the use of NMR spectroscopy or did not investigate SOM.

### 3.1. Atoms in the Composition of SOM Studied by NMR

The vast majority of the studies analyzed were studies using $^{13}$C (69.7%) and $^{1}$H NMR (21.1%) spectroscopy (Figure 3). Note that the number of studies exceeded the number of articles included. This is because an article could include studies on several SOM or soil atoms (Table S1). This is not surprising, as these atoms form the main framework of SOM. However, over the last decade, there has been a noticeable increase in the number of articles

investigating other atoms that are part of the SOM, and more and more articles have been published on the study of native soils and peat horizons. Authors are actively using NMR on the atoms $^{27}$Al, $^{29}$Si and $^{31}$P, which form the basis of the mineral matrix of soils.

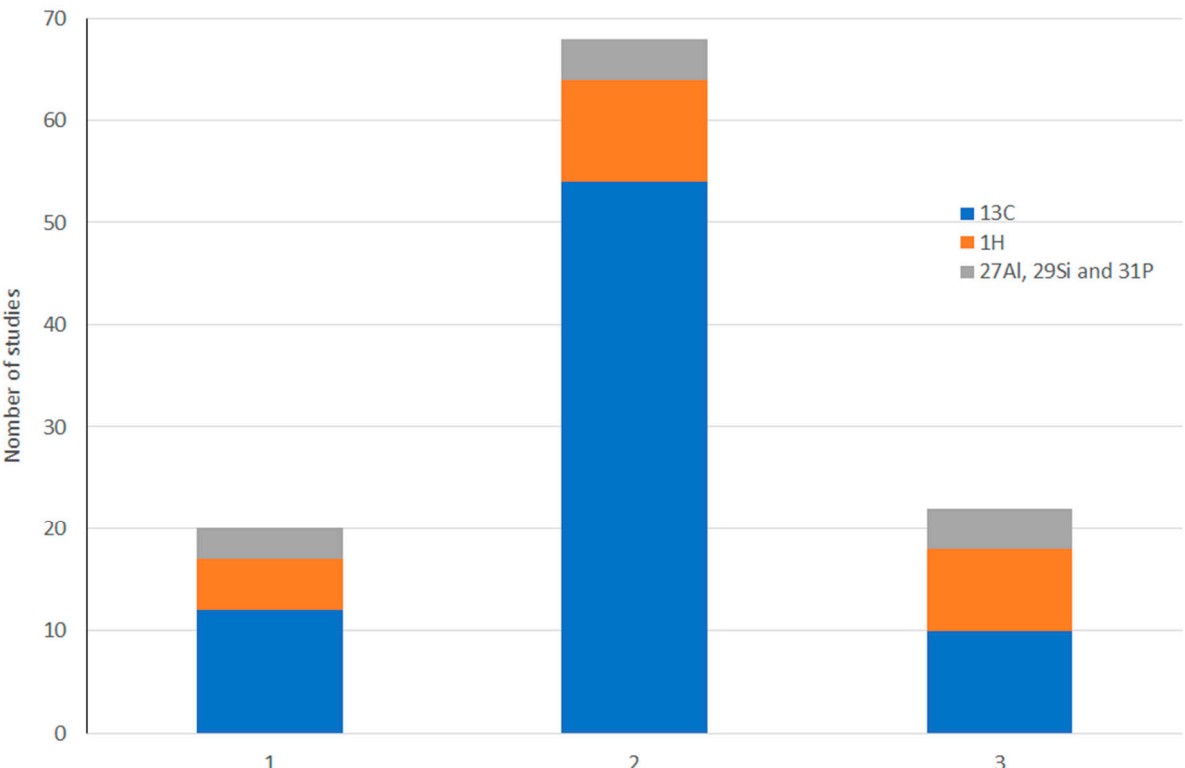

**Figure 3.** Distribution of the number of SOM studies by atom type using different NMR techniques: in solution (1), in solid phase (2) and both methods (3).

In the case of work with heavy nuclei, one often has to deal with the low natural content of isotopes with spin 1/2, such as $^{29}$Si, $^{15}$N and $^{13}$C, which also have a low gyromagnetic ratio as compared with protons, which leads to a lower Zeeman splitting of energy levels and, accordingly, to weak signals. As a result, the signal-to-noise ratio on the spectra is often unsatisfactory. Moreover, the spin-lattice relaxation at these nuclei in solids can be very long, reaching several minutes or more [23,24].

With the use of computers, it became possible to summarize a large number (up to several thousand or more) of spectra. This led to a significant increase in the sensitivity and the expansion of the application of this method for the study of nucleosensitive nuclei, such as $^{1}$H, $^{13}$C, $^{15}$N, $^{23}$Na, $^{27}$Al, $^{29}$Si and $^{31}$P in HS preparations [25,26]. Due to the significant advantages of pulsed techniques, new methods of hyperpolarization, including dynamic nuclear polarization and para-hydrogen-induced polarization, can further increase the sensitivity of the method by almost five orders of magnitude [27].

A certain methodological difficulty was the worse (compared to liquid techniques) resolution of solid-phase NMR spectra, especially in polycrystalline samples. Serious progress in the parametric interpretation of such spectra was achieved with the use of special mathematical processing of spectra based on the Fourier series (the so-called Fourier-transformation of spectra), which makes it possible to separate the superimposed singlet signals in the spectrum with a high accuracy and to estimate their integral intensity even with a poor signal-to-noise ratio [28].

### 3.2. NMR Spectroscopy Techniques

High-resolution NMR spectra of dissolved samples consist of narrow, well-resolved lines that correspond to magnetic nuclei in different chemical environments. The signal

intensities of the spectra are in some cases proportional to the number of magnetic nuclei in each structure, allowing for the qualitative and quantitative analysis of various mixtures (stereoisomers, tautomers, chemical reaction products, etc.) without prior calibration [1,2]. In addition, additional information about the structure and stereochemistry of molecules can be obtained from the superfine structure of spectral lines. The ability to detect all spectral lines allows for the identifying of intermediate products of chemical reactions (radicals, ions, ion-radicals, etc.), including short-lived ones, whose formation during chemical reactions was previously only postulated. This method makes it possible to study not only the kinetics but also the mechanisms of chemical reactions on the consumption and accumulation of initial, final and intermediate products [4,13,29].

The CP/MAS method helps to overcome the problems associated with the low concentration of atoms analyzed and the duration of signal accumulation. It works only in solids because it uses dipole interactions. Another advantage of the CP method is that the repetition rate of the experiment depends on the spin-lattice relaxation time of the nucleus from which the polarization is transferred, and the proton relaxation time is usually much faster than that of other nuclei with spin 1/2. This allows for more accumulations compared to a simple single-pulse experiment in which a rare nucleus is directly observed [20,30]. Thus, there is a double gain due to the increase in sensitivity and to the speeding up of the experiment.

The main interactions that make it difficult to obtain spectra with acceptable resolution are dipole–dipole interactions. They can be almost completely suppressed by rapid rotation of the sample around an axis tilted at an angle (54°44′) to the external magnetic field. This method is known as the "magic" angle rotation. A detailed description of the mathematical apparatus of most known solid-state exchange experiments using rotation at the "magic" angle was published in a review by Luz et al. [31]. However, this method works effectively only in the case of sample rotation with a frequency commensurate with the line width, which can reach tens or even hundreds of kHz. The first experiments used the CP/MAS scheme with a sample rotation frequency of 2–3 kHz. Over the past decade, the available speed limits have increased dramatically from 30 to more than 100 kHz [4]. This has been made possible by improvements in high-speed rotation equipment. Advances in the development of ways to increase sensitivity have transformed solid-phase NMR with ultrafast rotation with frequencies from 80 kHz and higher in an ultra-high-strength magnetic field into a practical tool for the structural chemistry of organic substances, including soil [3].

It should be noted that the use of the CP/MAS technique, although it increases the sensitivity of solid-phase NMR and significantly reduces the measurement time, slightly reduces the accuracy of the quantitative evaluation of carbon structures. The fact is that when using cross-polarization from $^1$H nuclei to $^{13}$C, the intensity of the $^{13}$C signal depends on the number and proximity of the surrounding $^1$H nuclei. This, of course, reduces the possibility of the absolute quantitative evaluation of different molecular structures in SOM preparations, but it does not hinder their comparative determination, provided that the cross-polarization time, proton relaxation time in the rotating coordinate system and relaxation time of individual $^{13}$C lines of NMR spectra are controlled when studying a series of samples [28]. This effect can be completely overcome only when recording spectra without cross-polarization, the so-called DP/MAS direct polarization method (which reduces sensitivity by a factor of about 4 and may require an increase in the time between scans due to the longer relaxation time in the $^{13}$C system) [20,32], or when taking spectra in solution [33]. Despite these difficulties, the CP/MAS method remains one of the most popular methods for soil studies due to its high sensitivity. This is confirmed by the analysis of the articles presented in this review; in 67.9% of the entries, only the solid-phase method was used; in 18.5%, the spectra were recorded in solution; and in 13.6%, both techniques were used (Figure 3).

### 3.3. Multidimensional NMR Techniques

Recently, two-dimensional (2D) and three-dimensional (3D) NMR spectroscopy have been used to analyze the molecular composition of HSs. In 22.2% of the articles analyzed, the authors used multidimensional NMR techniques. Two-dimensional heteronuclear spectra (hydrogen–carbon) are used more often. Hydroxy alkyl and alkyl protons are well identified, while the identification of the protons of aromatic fragments is difficult [34]. Since the imaging of heteronuclear NMR spectra of HSs is still possible only in solutions, the methods of preparation of these solutions are important for the quality of these spectra; nevertheless, the imaging of heteronuclear spectra allows for the verification of groups of structural fragments of HSs with greater validity than in single-nuclear (classical) versions of spectra [35].

The more commonly used heteronuclear single-quantum coherence (HSQC) method is a classical 2D NMR technique that correlates the chemical shifts of protons ($^1$H) with their directly bound $^{13}$C nuclei, yielding spectral peaks in the $^1$H-$^{13}$C spectral space. These peaks are further denoted by the $\delta_C/\delta_H$ ppm designation [36,37]. Two-dimensional $^1$H-$^{13}$C HSQC NMR spectra show different functionalities in the $^1$H-$^{13}$C spectral space (Figure 4).

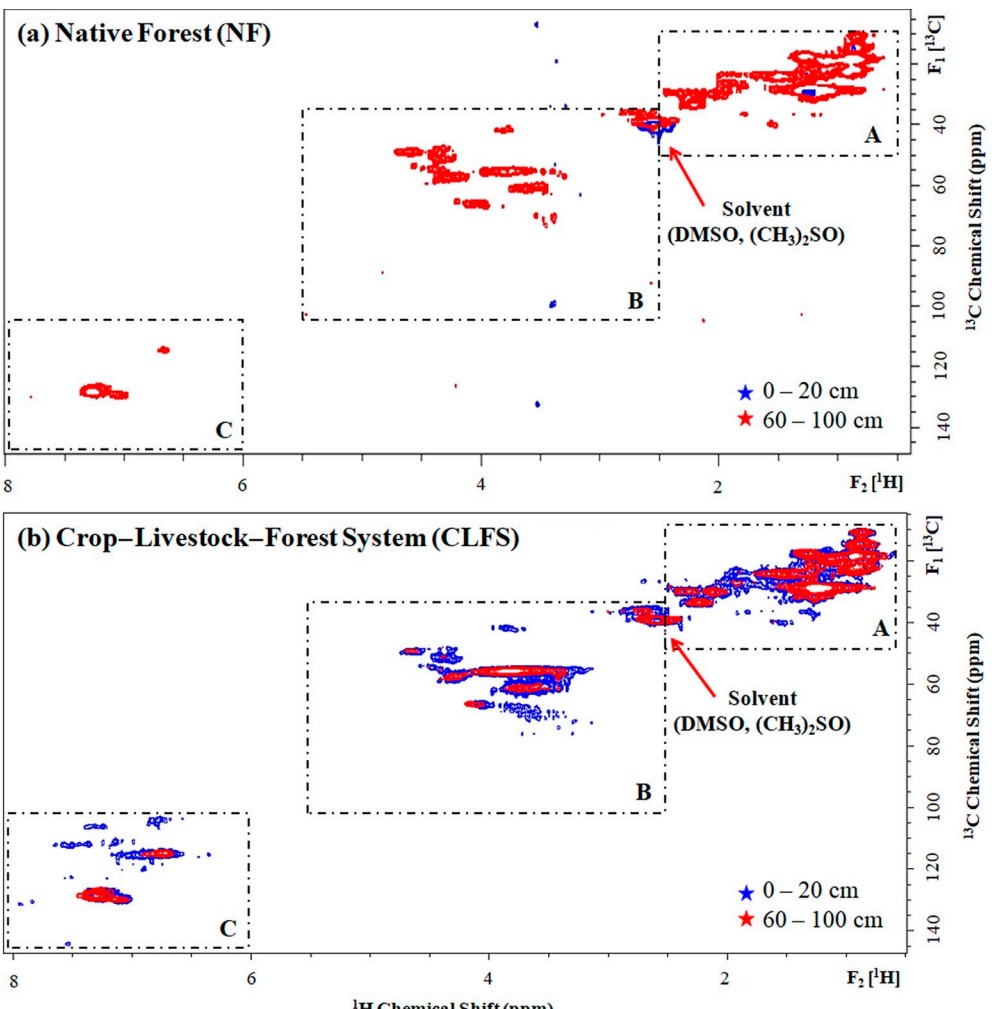

**Figure 4.** Two-dimensional $^1$H-$^{13}$C heteronuclear single-quantum coherence nuclear magnetic resonance (2D $^1$H-$^{13}$C HSQC NMR) spectra of HA samples extracted from (**a**) native forest (NF) and (**b**) crop–livestock–forest system (CLFS) at two different soil depths. Region labels correspond to the following: A = aliphatic; B = oxygenated aliphatic; and C = aromatic [38].

Various methyl (-CH$_3$), methylene (-CH$_2$-) and methine (=CH-) groups ($\delta_C/\delta_H$: 10–50/0.5–2.5 ppm) are observed in the aliphatic region. Resonances from functionalized carbon chains (substituted with O- or other hetero-element-containing groups), as well as carbohydrate resonances ($\delta_C/\delta_H$: 35–110/2.5–5.5 ppm), are present in the oxygen-containing aliphatic region. Aromatic functional groups resonate in the weak field ($\delta_C/\delta_H$: 110–150/6.0–8.0 ppm) [38,39]. The hydroxyalkyl and alkyl protons are well identified, while there are significant difficulties in identifying the protons of aromatic fragments [34]. Since the imaging of heteronuclear NMR spectra of SOM is still possible only in solutions, the methods of preparation of these solutions are important for the quality of these spectra; nevertheless, the imaging of heteronuclear spectra allows us to verify the groups of structural fragments of SOM with greater validity than in the uninuclear (classical) versions of the spectra [35,40].

The use of 2D $^1$H-$^{13}$C NMR spectroscopy provides more detailed data than $^{13}$C CP/MAS or $^1$H NMR spectroscopy alone and thus represents a reliable tool for the identification of individual structural fragments in HA molecules. The use of this method is limited by the high labor-intensiveness and complexity of the data interpretation; therefore, this method is mainly used for various natural polymers [41], including SOM. Further study of natural organic substances by this method will open new views on the formation of organic acids in various environments as well as on their interaction with various chemical structures.

### 3.4. Interpretation of the NMR Spectra

The application of $^{13}$C NMR spectroscopy in the study of SOM has significantly expanded our understanding of its composition and structure and deepened our understanding of the mechanisms of humification and transformation [3,42–45].

The main indicator characterizing the signal position (from magnetic nuclei in different chemical environments) in the NMR spectra is the value of the chemical shift, expressed in ppm (parts per million), from the zero-value correlated to the NMR signal of a reference compound such as tetramethylsilane. Such tables are widely known and have been published [13,46,47]. It should be noted that due to the strong overlap of the peaks, the assignment of signals in them (Figure 3) is only possible according to the chemical shift ranges, according to the position of resonances of atoms with a similar chemical environment [48]. Since the spectra often show a small variation in chemical shift values depending on the spatial configuration of the surroundings of certain molecular fragments and chemical shift anisotropy, the chemical shift ranges in which the signals of molecular fragments and functional groups are observed are often used in the quantitative treatment of SOM spectra [49–51]. For the same reason, the start and end values in these ranges vary somewhat between authors (Table 1).

**Table 1.** Chemical shifts of C$^{13}$ atoms of SOM molecular fragments.

| Chemical Shift, ppm | | Type of Molecular Fragments |
|---|---|---|
| **Range Start** | **Range End** | |
| 0 | 45–52 | C,H-substituted aliphatic fragments |
| 45–47 | 60–2 | methoxyl and O-, N-substituted aliphatic fragments |
| 52–62 | 95–120 | aliphatic fragments, twice-substituted heteroatoms (including carbohydrates) and methine carbon of esters and ethers |
| 95–120 | 140–145 | C,H-substituted aromatic fragments |
| 140–145 | 160–170 | O,N-substituted aromatic fragments |
| 160–170 | 183–185 | carboxylic groups, amides and their derivatives |
| 183 | 190 | quinone groups |
| 185–190 | 204–220 | aldehyde and ketone groups |

The analysis of the selected articles showed that in 77.8% of them, the authors use the indicated ranges of chemical shifts to decipher the obtained $^{13}$C NMR spectra (Figure 5).

The remaining articles, as a rule, are devoted to studies of other atoms in the structure of SOM or the authors limited themselves to the interpretation of individual peaks (Table S1).

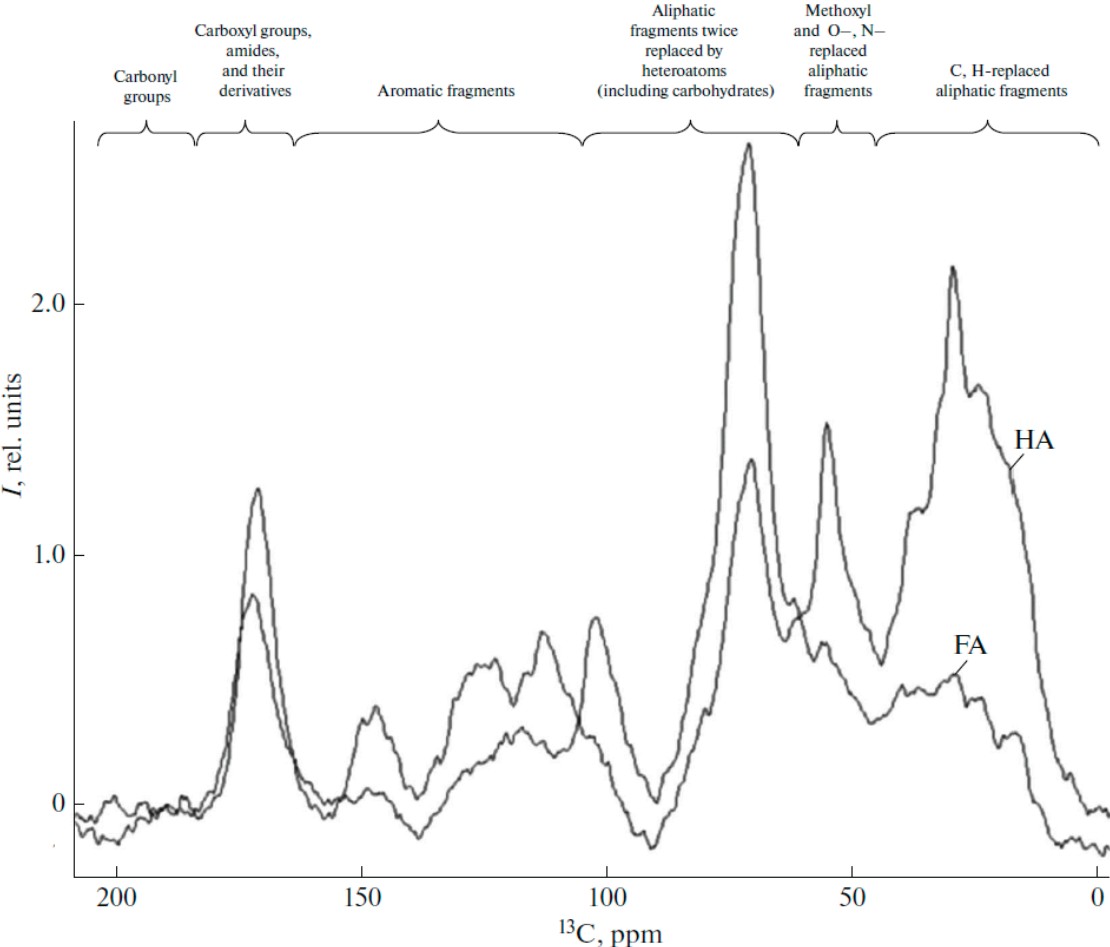

**Figure 5.** Typical solid-phase $^{13}$C NMR spectra (CP/MAS technique) of HSs, by the example of humic acid (HA) and fulvic acid (FA) preparations from the O (folic) horizon of Retisols [51].

As follows from Figure 5, the intensity maxima are observed in the range of unsubstituted aliphatic fragments; all $^{13}$C NMR spectra have one relatively sharp peak concentrated in the 30 ppm region, which can be attributed to methylene carbon atoms in the α, β, δ and ε positions from the end methyl groups (15 ppm) in the alkyl chains [52,53]. These methylene carbon atoms, as suggested by some researchers, may be the result of accumulation of wax resins, lipids and suberin structures from plants [20,49]. Also, all spectra have a signal in the 47–60 ppm range, which some authors attribute to methoxyl groups due to the presence of lignin fragments and syringyl and guaiacylpropane units [54,55]. However, signals from carbon atoms in the α-position in polypeptides (-C(O)-C*(R)H-NH-)$_n$ can also be observed in this chemical shift region [49].

A very intense peak at 71 ppm in $^{13}$C NMR spectra is characteristic of the CH(O)-groups of ring carbon atoms in carbohydrates [54,55]. The signal in this region of the spectrum extends from 64 to 90 ppm, which probably originates from various HC-OH groups of cellulose or other carbohydrate fragments [49]. In addition to this peak, additional signals of carbohydrate structures have been identified in most $^{13}$C NMR spectra of HA and FA. A weak signal around 62 ppm is typical for the CH$_2$O groups of hexoses from polysaccharide fragments [56]. The presence of a weakly intense signal at 101 ppm, which is representative of anomeric (semiacetal) carbon atoms, also confirms the presence of

carbohydrate fragments in the structure of HA and FA. The most intense signals in the "carbohydrate" region were observed for FA.

In the region of aromatic fragments, the signal within 108–144 ppm can be given by unsubstituted and/or alkyl-substituted aromatic carbon atoms. The peak at 147–149 ppm is typical in the spectra of lignin structural units, and it is attributed to the oxygen-substituted carbon atoms of the aromatic rings syringyl and guaiacylpropane units [49,57]. NMR spectra of HA have more intense signals in this region compared to FA (Figure 5).

In the carboxyl group region (164–183 ppm), there is a maximum at 171–173 ppm, largely attributed to the carbon of carboxyl groups [58], but it can also belong to the carbonyl group of amides and polypeptides [49]. Carbon atoms of quinone fragments and carbonyl groups of aldehydes and ketones contribute to very weak signals in the 183–190 and 190–204 ppm regions, respectively.

A number of integral indices (descriptors) are often used to quantitatively characterize the structure of HSs. First, these are the ratios of carbon contained in various structural components. The most important characteristic is the ratio of alkyl and aromatic fragments in the HSs, i.e., AR/AL [13] or $\Sigma C_{Ar}/\Sigma C_{AL}$ [50], referred to by the authors as the degree of aromaticity; aromaticity can also be calculated using the formula proposed by Liang et al. [59], i.e., AR/(AR + AL), %, where the authors sum the signals from aromatic structures over the regions of 106–170 ppm and of aliphatic and aliphatic carbon atoms over the regions of 0–170 ppm. Both approaches give a very high correlation between the values obtained in assessing the aromaticity of HA. An index characterizing the total fraction of unoxidized carbon atoms, i.e., substituted by hydrogen or carbon atoms only, which allows for an indirect assessment of the amphiphilic properties of HSs, can be calculated using the formula $AL_{H,R} + Ar_{H,R} = ((0–47 \text{ ppm}) + (108–144 \text{ ppm}))$ [42]. The C,H-alkyl/O,N-alkyl ratio, which reflects the degree of organic matter decomposition, is calculated from the major chemical shift ranges corresponding to the alkyl (0–47 ppm) and O,N-alkyl (47–108 ppm), fragments [32,60].

The choice of the used integral indices is always determined by the goal and objectives of the study, but one should keep in mind that when interpreting them and comparing them with each other, one should take into account the error of the calculated values. In addition, authors very often use different parameters for the acquisition of NMR spectra (contact time, delay time, spinning frequency of the rotor for solid-phase technology, etc.). The analysis of the articles selected in this review showed that in 77.8% of the cases, the researchers did not choose the conditions but used parameters taken from other publications. Therefore, the lack of control over the choice of parameters of the NMR technique used can lead to significant errors in the comparative analysis of the results obtained.

### 3.5. NMR Investigations of SOM Pools

Using NMR spectroscopy, the processes of transformation and establishment of the SOM system over time under the action of various factors (agricultural use, burying, primary succession, fires, etc.) on soils were studied. The articles analyzed cover a wide range of different SOM pools, from high-molecular-weight biopolymers (HS fractions, lignin, dissolved organic matter (DOM)) to single organic substances (polycyclic aromatic hydrocarbons, pesticides, antibiotics, etc.). Most articles (67.9%) are devoted to the study of the molecular composition of HS fractions—HA, FA, humin or unfractionated HSs (Figure 6). Much less frequently, native mineral and/or organic (peat) soil horizons were studied by NMR (35.8%).

It has been established that the molecular composition of HSs varies over time due to the different composition of the litterfall and the rate of its transformation [61,62]. At the same time, the content of aromatic fragments increases in chronoseries soils on quarry dumps compared to aliphatic ones, which is especially pronounced in the case of controlled reclamation. At the same time, the inflow of fresh litter fractions into the surface horizons regulates the composition and content of aliphatic components. In the chronoseries of different-aged podzols formed on sand open-cast dumps, the HA composition of the

topsoils contains more aliphatic and alkylaromatic fragments [39] than in HSs of eluvial and illuvial horizons. The molecular composition of the HSs of young podzols strongly depends on the type of plant material entering the soil [63]. A close relationship between the molecular composition of HSs and the composition of humification precursors entering the soil was also observed for Chernozemic soils [64], which indicates the differentiation of soil organoprofiles on a vertical scale in terms of the molecular composition of HSs. The dynamics of HA molecular composition was noted not only for chronosequences but also for climatic series of soils in the altitudinal gradient [65]. In both cases, it is associated with changes in the composition of plant residues entering the soil. At the same time, the age of the SOC, determined by the type of horizon from which it was extracted, is largely related to its structural composition, at least for gray soils and Cambisols [65].

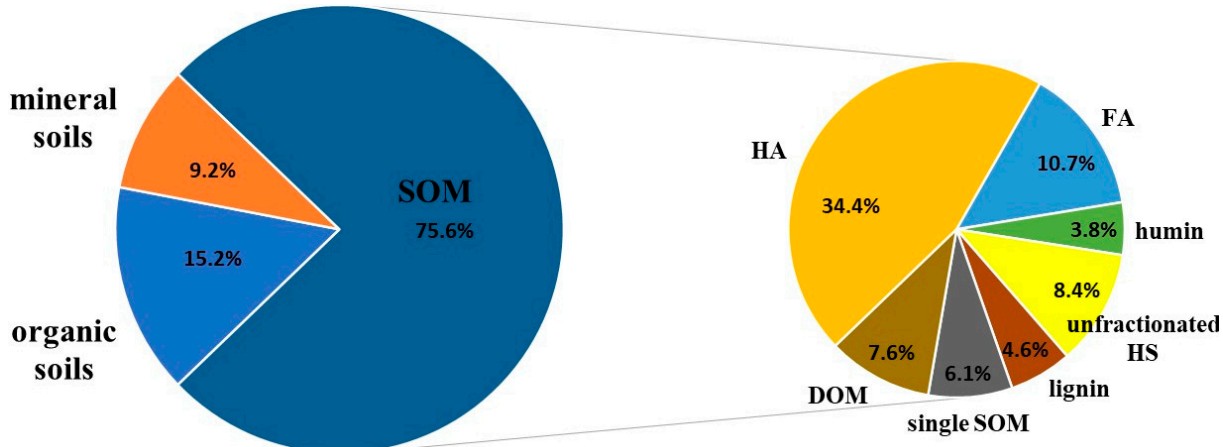

**Figure 6.** Distribution of NMR studies by object.

Significant changes in the molecular structure of HA during soil burying have been established. By comparing the $^{13}$C NMR spectra of HA of modern and buried chestnut soils, it was established that the HA spectra of modern soils are characterized by the greatest intensity of signals in the region of aliphatic carbon, carbon of carboxyl groups and the presence of two peaks of insignificant intensity in the region of O,N-substituted aromatic fragments, which disappear in the preparations of HA of paleosoils [66]. Changes in the structural composition of the carbon skeleton have been noted by different authors for buried soils of various regions. In the middle taiga, the relict HSs sharply differ from the modern ones: in addition to the 17% increase in the aromaticity of relict HSs, the proportion of oxidized fragments of both an aliphatic nature (carbonyl and carboxyl functional groups) and an aromatic nature (phenolic and quinone molecular fragments) increases. In addition, a smoothing of all peaks in the $^{13}$C NMR spectrum of relict HA was observed, which indicates the acquisition of thermodynamic stability and homogeneity of the carbon skeleton of the HA of paleopedorelicts [67]. An increase in the degree of aromaticity of HA was also observed in three soils with different burial ages in the forest-steppe zone; the highest was in soils with a lower burial age (400 years), and the lowest was in soils with a longer burial time (1000 and 2000 years) [8]. An increase in the degree of aromaticity of HSs of buried soils was also noted for polar ecosystems [68].

The application of $^{13}$C NMR spectroscopy proved to be informative in the study of soils subjected to wildfires. Due to the fact that this phenomenon leads to the widespread transformation of organic matter and an increase in black carbon pools [69–71], a detailed study of the molecular structure of SOM subjected to pyrogenesis has been carried out [72–74]. In particular, a loss of alkyl components and a relative increase in the content of aromatic fragments was observed, which indicates the apparent stabilization of organic matter molecules [75].

In addition to studying the component composition of SOM or HS preparations, [13]C NMR spectroscopy is actively used to characterize the initial humus-forming substances, also known as humification precursors [76]. It has been noted that lignin components and alkyl fragments account for a significant proportion of the composition of wood remnants [77]. At the same time, [13]C NMR spectroscopy allows us to separate the components of organic matter composition inherited from parent materials containing lignite and those formed during soil formation [78]. It is interesting that changes in the structural composition of the understory occur in relatively rapid scenarios, particularly in tree clear-cutting, which leads to an increase in aromatic fragments [79], which we have also noted below for post-fire changes in soil cover. Thus, the structural composition of forest litter is not completely conservative and can change rapidly. At the initial stages of humification, the composition of forest litter is dominated by alkyl fragments. The use of NMR spectroscopy in combination with other methods makes it possible to identify the contribution of individual plant residues to humification [80,81]. The relationship between the structural composition of organic matter and the density of the corresponding fractions and the degree of decomposition of detritic fractions of organic matter is also quite interesting [82].

The application of organic fertilizers (manure, peat, biochars, humates of various origins, etc.) to agricultural soils increases the content of SOM, increasing soil stability by improving its physical, chemical and biological properties [2,83]. Organic fertilizers in the soil are biologically transformed through mineralization. The remaining detritus is then transformed into HSs by humification. This increases the share of aromatic structures in the composition of SOM, which makes the HSs resistant to further decomposition and potential carbon accumulation in the soil [84].

The agricultural use of soils changes not only the concentrations of SOM in fine earth but also affects the molecular composition of HSs [51]. Changes in hydrothermal conditions and the composition of incoming plant residues affect the rate of humification and its products. A comparison of the molecular composition of HSs isolated from mature and arable Retisols using [13]C NMR spectroscopic analyses showed that soil development increases the proportion of aromatic components and decreases the content of carboxyl and ester groups in the HS structure. Increasing the degree of hydromorphism of Retisols leads to the enrichment of HSs with aliphatic fragments [51]. Regular, long-term organic additions enhance the aromatic characteristics of HSs, which can improve soil functionality. However, short-term structural improvements are only achievable when the applied material is rich in aromatic compounds [81].

Studies of the HSs of arable and mature Chernozems showed that the involvement of mature Chernozems in arable farming leads to a decrease in aliphatic fragments in the HS structure. The AR/AL ratio was lower in HSs from arable soils samples, which the authors explain by the microbiological destruction of the hydrolyzable aliphatic part of the HSs due to the traditional tillage system (plowing) and a lack of conditions for the accumulation and decomposition of plant residues [85]. Similar results were obtained in other works on the study of SOM of arable Chernozems. At the same time, the researchers note a decrease in the number of aliphatic chains and an increase in the content of aromatic structures and carboxyl groups in the composition of the HSs [86,87].

Increasing the supply and stability of SOM through the application of biochar and lignite waste to soils pays both agronomic and environmental benefits and represents a win–win solution to the growing challenges of food security and climate change [88]. [13]C NMR spectra of HAs isolated from coal are characterized by the highest spectral density in the range of aromatic carbon ratios (100–165 ppm) and account for more than 50% of the total spectral area [89]. Coal waste derivatives have high hydrophobic properties [90], which is a predisposition to the increased sequestration of organic carbon in soil. The application of coal waste to soil can contribute to a reduction in $CO_2$ emissions from arable soils.

At present, fairly satisfactory [13]C NMR spectra of the structural components of organic matter have been obtained directly from soil samples [91]. However, obtaining high-quality spectra requires quite a long time of continuous operation of a pulsed NMR radio

spectrometer. Because of the relatively low organic carbon content of soil samples, to obtain a spectrum suitable for interpretation and mathematical processing, it is necessary to accumulate 30–50 thousand scans on each sample, which takes more than 30 or more hours of instrument operation [92]. Laboratory practice shows that taking high-quality solid-phase $^{13}$C NMR spectra is difficult in the presence of a large number of paramagnetic centers due, for example, to the presence of iron, manganese, etc., ions in the sample. $^{13}$C NMR is optimally used for organogenic (e.g., peaty, histic, folic) soil horizons or densimetrically extracted SOM fractions when the sample does not contain large amounts of iron or manganese salts. In this case, additional de-salting by dialysis, resuspension or hydrofluoric acid treatment is often performed for HS preparations [28,93]. The analysis of the reviewed articles shows that almost all the authors investigating native peat litter or mineral horizons of soils pre-treat the samples with HF solution prior to $^{13}$CNMR analysis. This procedure is suitable for increasing the concentration of organic carbon and favours the elimination of carbonates and paramagnetic compounds that can cause broadened resonances and a reduced signal intensity in the spectrum. However, one should keep in mind that the additional treatment of HS preparations and soil samples with acid and alkali solutions will disrupt the nativity of the drug, leading to hydrolysis and the detachment of peripheral fragments of HSs included in the structure of their macromolecules.

The above-mentioned obvious advantages have led to NMR spectroscopy becoming one of the most frequently used methods for studying the molecular structure of SOMs, which is primarily due to its high informative value and relative ease of obtaining results [2,94–96]. Biochemists have obtained a unique direct non-destructive solid-phase method to study the carbon skeleton of complex organic macromolecules, which also allows for studying the processes of their interaction with each other, with mineral particles and with low-molecular-weight xenobiotics like pesticides and antibiotics. Since that time, there has been a veritable explosion of applied research that has deepened the fundamental understanding of SOM, the processes of its transformation and the nature of its interaction with toxicants of various nature [97].

## 4. EPR Spectroscopy

The EPR method has been successfully used in the study of SOM over the past decades since it is highly sensitive and provides extensive information on the molecular structure of compounds containing unpaired electrons. However, it is an order of magnitude less commonly used to study SOM than NMR (Figure 2). This is most likely due to both the specificity of the information obtained and the limited skills of researchers in working with the EPR method.

### 4.1. The Nature of SOM Paramagnetism

There are different opinions in the literature on the nature of the paramagnetism of HSs. Some researchers attribute it to the presence of FRs stabilized by a rigid polymeric matrix of organic matter (Figure 7) and explain the paramagnetism of HSs by the generation of unpaired π-electrons due to the formation of defects in the polyconjugation systems caused by the excess energy that takes place during the formation of the solid phase, with the predominant role of the synergistic interaction between aromatic structures (and even individual rings) and hydrogen bonds formed by different functional groups [98,99]. According to Lishtvan et al. [100], the principal difference between the nature of the paramagnetism of highly dispersed structures and classical FRs is that in the latter, the unpaired electron belongs to the outer atomic or molecular orbital of a single (isolated) molecular fragment, while in highly dispersed HS structures, it is delocalized over several molecular fragments. In highly dispersed particles, there is no migration of the unpaired electron as a particle but a delocalization of the spin density characteristic of low-dimensional structures. This circumstance accounts for both the unusual stability of the paramagnetism of HSs (unpaired electrons do not recombine in the liquid phase) and the sensitivity of EPR spectra to the chemical structure and size of highly dispersed particles.

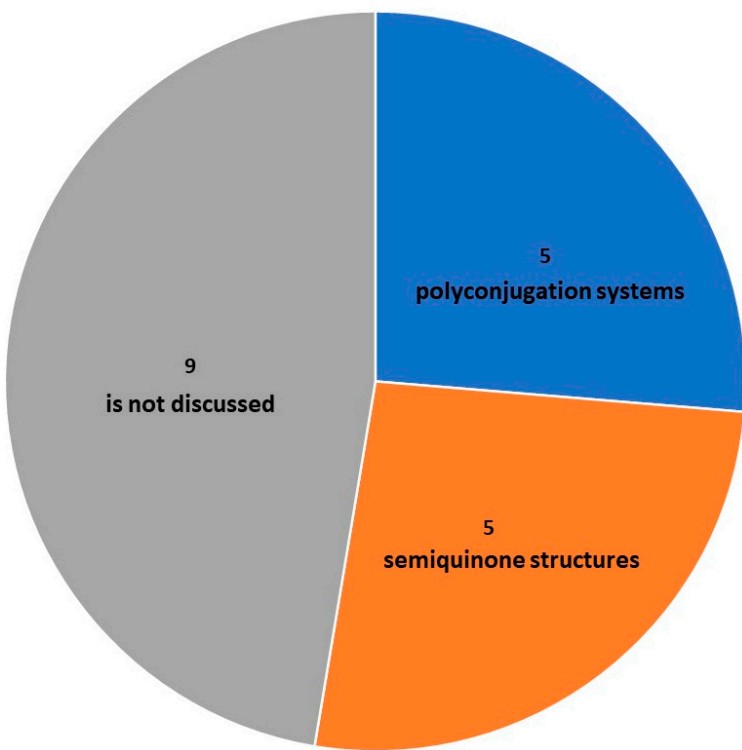

**Figure 7.** Distribution of views of the authors of the selected articles on the nature of SOM paramagnetism.

There is also a fairly widespread belief (in half of the papers where authors discussed the nature of SOM paramagnetism) that the paramagnetism of HSs is mainly due to semiquinone radicals and is justified by the absence of significant differences in the shape of the EPR spectra of the original samples and those treated with alkaline solutions, as well as the positive correlation between the signal intensity and the content of quinoid groups [101,102]. The formation of FRs during enzymatic oxidation of HSs containing phenolic fragments has been noted by many researchers. Equation (1) can be used to describe the reaction mechanism [103]:

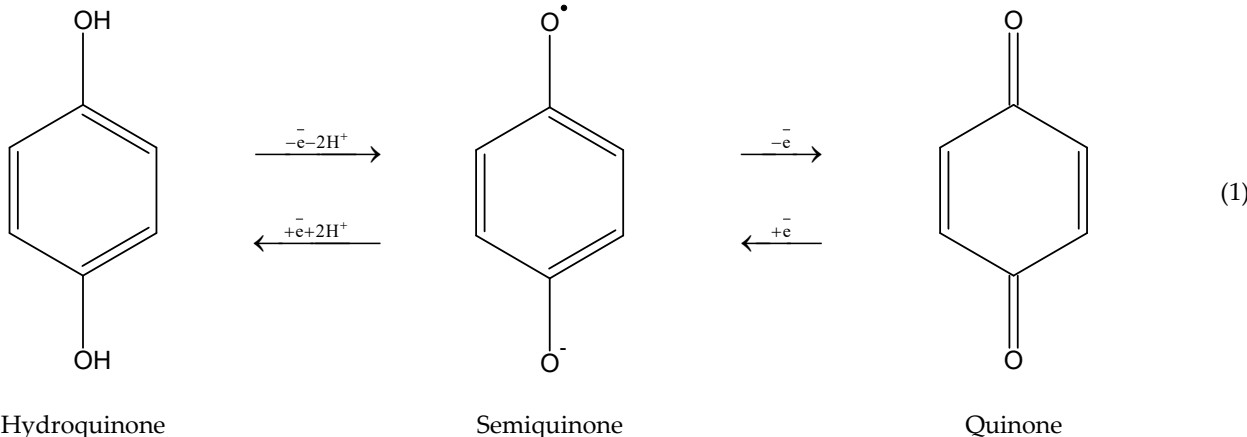

(1)

The redox process is a stepwise reversible oxidation of hydroquinone through the formation of semiquinone, a particle containing FR.

However, these facts leave open the question of the source of the HA paramagnetism: semiquinone radicals or polyconjugated fragments. In addition, this concept is difficult to agree with the fact of precipitation of HA by mineral acids, because it is known that semiquinone radicals are stable only in the ion-radical form, so they are generated and stored for a long time only at alkaline pH values, while preparations of HA have an

acidic reaction. The results obtained by Kurochkina et al. [104] indicate that in an alkaline environment, the EPR signal is indeed mainly due to semiquinone radicals. In an acidic medium, they disappear, so the paramagnetism of the H form is mainly due to FRs, which are characteristic of polyconjugated structures [105,106].

All compounds that contain unpaired electrons are divided into two groups. In the first group, the unpaired electrons are bound either to the whole molecule or to most of it. Such electrons travel along strongly delocalized molecular orbitals and determine the activity of the groups of atoms that are part of the molecule's structure. The study of these unpaired delocalized electrons is extremely important in deciphering the specifics of the processes that take place through the stage of FR formation and in understanding the mechanism of polymerization reactions or the formation of intermediates of various chemical reactions. The second group of substances can include compounds in which the unpaired electron is bound to a single atom rather than moving along molecular orbitals linking several atoms [106]. Therefore, the use of EPR in the study of biological and biochemical systems that contain atoms of paramagnetic metals or their ions can provide valuable information about the degree of their oxidation and the nature of the bonds of the atoms under study.

It is well known that HSs contain conjugated aromatic structures in their molecular composition. A distinctive feature of such structures is the overlapping of the $2p_z$-orbitals of neighboring atoms with the formation of molecular $\pi$-orbitals, which allows for the delocalization of electrons throughout the molecular system. And, if the unpaired electron is in the $\pi$-orbital, then such a radical is called a $\pi$-electron radical. Most organic FRs can be referred to as this class. Only a small number of radicals can have an unpaired electron localized on the $\sigma$-orbital. As a rule, such an orbital is located in the place in a molecule at which there is not enough atom to saturate the valence. These radicals are usually bound to transition metal atoms—Cu, Mn and Fe—and their amount in the atom, as well as their energy, changes with the change in the oxidation degree of the atom to which they are bound [107]. Therefore, studies of the paramagnetic properties of HSs containing ions of such metals can provide the necessary information about the nature of the bonds of the studied atom with HS molecules.

A large number of works have been devoted to the elucidation of the nature and properties of stable FRs. Many authors tend to believe that FR activity is a fundamental property of HSs [108,109]. However, to date, the question remains open as to the reasons for the birth of FRs in polysynthesized fragments of HSs. The literature data indicate a number of possible reasons, which can be reduced to two main ones: firstly, the conditions of the formation of polymer structures in the solid matrix and, secondly, the peculiarities of the electronic structure of the HS molecules.

The formation of FRs in polyconjugated systems can be caused not only by the specific structure of their molecules but also by the intermolecular interactions of polyconjugated systems in the solid state [106]. Taking into account the high correlation between the energy spectrum of $\pi$-electrons and the geometric structure of the macromolecules of polysoluble systems, the authors linked the molecular interactions in them with specific and strong collective interactions of delocalized $\pi$-electrons along the polymer chains. The formation of associates in which the polysynthetic systems of HS molecules are stacked on top of each other is energetically more advantageous. Specific physical conditions may arise at the boundary of associates, and conjugated bonds that get into the boundary regions may become "defective" bonds as a result of the release of elastic energy due to the nonequilibrium dynamic process of the formation of the supramolecular structure of organic matter during the formation of the solid phase. A consequence of the proposed model [110] is the presence of two types of magnetically non-equivalent FRs. When the "defective" bond is broken, two unpaired $\pi$-electrons are formed, one of which (wide signal) belongs to the fragment outside the association and the other (narrow signal) belongs inside. The magnetic non-equivalence of the formed unpaired electrons is due to the difference in their states inside and outside the association. The close spatial arrangement of both types of FRs can lead to their interaction. In the case of synthetic polysynchronous systems,

it manifests itself in the inversion of the phase of the narrow signal at high microwave power levels and small modulation amplitudes. In the case of HA, it manifests itself in a resolution much better than expected for a superposition of two lines with given widths. It is believed that the formation of HS associates can occur as a result of the direct interaction of functional groups as well as through water molecules and multivalent ions.

### 4.2. Investigations of the SOM Pools and Parameters of the EPR Spectra

Like NMR, EPR is used to study extractable SOM pools as well as native soil samples. In the vast majority of the selected studies (85.7%), the authors investigate the content and properties of FRs in the structure of HS fractions (HA, FA and humin) or unfractionated HSs. Other pools of FRs were completely absent in the selected articles. Only three articles out of the nineteen selected (Table S2) were dedicated to the study of the paramagnetic properties of mineral and/or organic soil samples, representing 14.8% of the total number of studies (Figure 8).

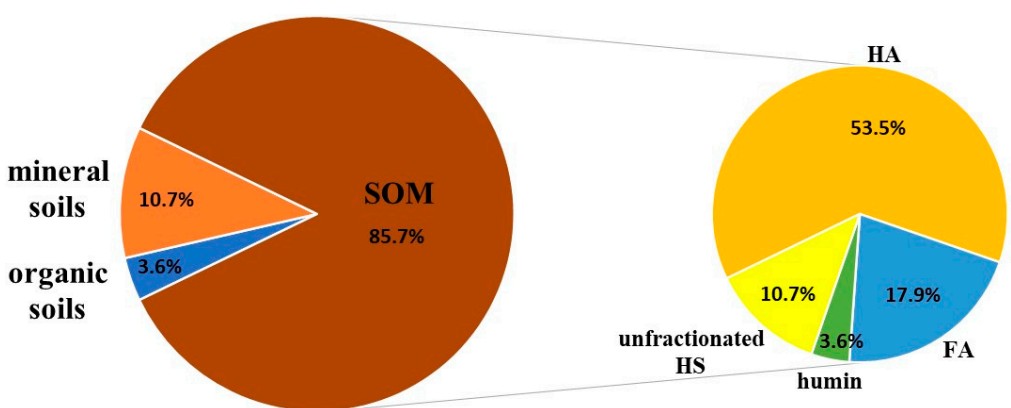

**Figure 8.** Distribution of EPR studies by object.

There is no consensus in the literature on the change in the concentration of FRs in HSs during soil development due to the small number of works on the SOM of arable soils (Figure 9).

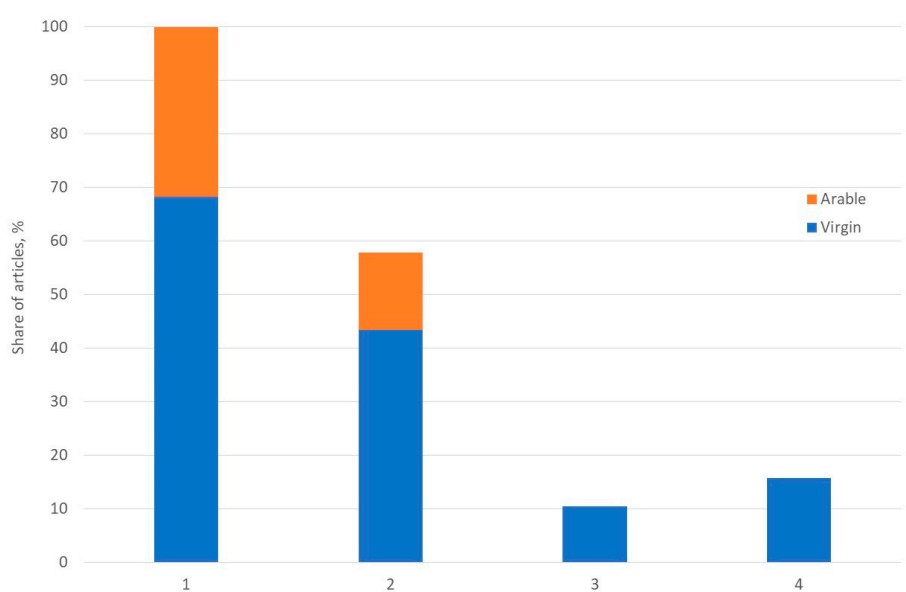

**Figure 9.** Distribution of articles using different parameters of EPR spectra in their analyses in the study of virgin and arable SOM: FR concentration (1), g-factor value (2), spectral shape (3) and peak-to-peak line width (4).

Thus, the agricultural use of Retisols differently affects the FR content in HA and FA. The FR content in HA from the arable horizon is approximately the same as the average FR concentration in HA from the upper horizons of native Retisols. In contrast to HA, the FA of arable Retisols is less thermodynamically stable and more amenable to microbiological degradation. This, in turn, causes a more intensive transformation of FA molecules during cultivation, as a result of which the concentration of FRs sharply decreases to trace amounts [106]. According to some researchers, the concentration of paramagnetic centers in HA is the inverse of their biothermodynamic stability and humification depth. During the tillage use of Chernozems, the paramagnetic activity of HA decreases [105]. Other authors believe that the involvement of some crop residues in HSs can lead to an increase in the number of FRs [109,111]. In this case, a large number of FRs in the HS structure act as microbial electron acceptors and facilitate the sequestration of organic carbon in the composting process. Therefore, improving the composting microenvironment and promoting microbial anabolism are important for improving the quality of the HSs during composting fermentation [112]. The addition of iron salts enhances the synergistic quinone oxidation–reduction cycle of iron reduction and promotes FR formation during humification in composting [113].

A characterization of HA extracted from Latosols (Brazil) under conditions of integrated agricultural systems (crop, livestock and their combinations), pastures and natural forests showed that the HA humification index increased with depth in all the soils studied. This was determined by complex structures richer in FRs of a semiquinone type. At the same time, the HA from soils under agricultural production had a higher aromaticity profile than the HA of soils from pastures and natural forests, which suggests a longer lifetime of carbon compounds [114].

A study of the fate of environmentally persistent FRs in agricultural soils after biochar application has shown the influence of two main factors: the method of cultivation and the content of transition metals in soils. Cultivation and plowing accelerate the decomposition of all types of FRs, both oxygen- and carbon-centered, regardless of their initial content or the level of biochar application. The content of transition metals in soils determines the type of FRs. Although initial FRs from biochar are predominantly of a carbon-centered origin, transition metals in soil or applied with fertilizers lead to a slow transformation/formation of oxygen-centered FRs [115,116].

Authors often limit their studies only to estimating FR concentrations in the composition of SOM or soils (Figure 9). Another 57.9% determine and discuss g-factor values in their articles. At the same time, they rarely discuss and attach importance to the specificity of EPR spectral lines, their shape and their width. The inability or unwillingness of authors to extract qualitative information from the use of EPR spectroscopy contributes to its limitation compared to NMR spectroscopy.

## 5. Conclusions

A wide range of articles devoted to the use of NMR and ESR spectroscopy methods for studying organic matter of mature and agricultural soils were analyzed in this paper. It has been shown that both solid- and liquid-phase NMR spectroscopy are widely used to investigate the structure of SOM. Solution-state NMR is very informative when studying soluble SOM components, especially those with a low molecular weight. However, compared to solid-state NMR, the use of solutions has some disadvantages when applied to SOM: (1) Some samples or fractions of SOM are insoluble. (2) Solid-state NMR promotes a much higher concentration of $^{13}$C atoms in the sample than when shooting in solutions, which increases the signal and saves analysis time. Attempting to increase the SOM concentration in the solution to achieve a strong signal can promote aggregation, resulting in a lower sensitivity, lower resolution and loss of information about the structure. The conversion of samples to a solution, on the other hand, may result in the inability to further analyze them by other methods. (3) It is much easier and simpler to detect non-protonized carbon atoms using solid-state NMR. (4) In solution NMR, rapid rotation leads to anisotropic

interactions, whereas in solid-state NMR, these anisotropic interactions can be manipulated by specially designed pulse sequences. (5) Solid-state methods can identify non-uniform and heterogeneous regions within structures. Given the above limitations associated with solution NMR spectroscopy when applied to SOM, their spectra should be interpreted with caution.

Combining information obtained by two- and three-dimensional NMR spectroscopies with other physicochemical methods (pyrolysis, chromatography, mass spectrometry, etc.) may offer a much more powerful approach to unraveling the mysterious composition of SOM than was previously shown using one-dimensional analogs of these methods.

The advantages of EPR over other methods of experimental biochemistry are significant. It has a very high sensitivity and, without disturbing the structure of the compounds under study, gives a variety of original information about the structure of substances containing FRs, which are usually understood as molecules, fragments of molecules or individual atoms in their structure that have a free (unpaired) electron. However, in order to make it popular with SOM researchers, a broader approach to extracting the information that the EPR method can provide is needed. By studying the FR structure of various biopolymers in nature, it is possible to assess their reactivity as well as obtain information about all kinds of their transformations in different environments. In this connection, a detailed study of the molecular structure and properties of FRs in SOM and soils is a promising and topical task in solving the problems of studying the formation and transformation of SOM.

This scoping review has some limitations. To make our review more feasible, we limited ourselves to the period from 1996 to 1 March 2024. In addition, a large number of articles were published in Chinese, Spanish, Russian and other languages. Therefore, our results can only be generalized to English-language research articles. The conclusions and suggestions presented are based on the literature over a broad time period but may be slightly biased by the research questions and selection criteria (Figure 1). The proposed framework was derived empirically and may not be representative of other methods for investigating SOM, organics from other environments or other time periods beyond our selection. In this sense, this study is a first step in demonstrating the need to contextualize the design of studies on the properties of SOM and the choice of appropriate methods.

**Supplementary Materials:** The following supporting information can be downloaded at: https://www.mdpi.com/article/10.3390/agronomy14051003/s1, Table S1. Included articles focusing on the use of NMR spectroscopy and results of individual sources of evidence. Table S2. Included articles focusing on the use of EPR spectroscopy and results of individual sources of evidence.

**Author Contributions:** Conceptualization, E.L. and E.A.; methodology, E.L.; validation, E.L. and E.A.; formal analysis, E.L. and E.A.; investigation, E.L.; resources, E.L.; data curation, E.L. and E.A.; writing—original draft preparation, E.L.; writing—review and editing, E.L. and E.A.; visualization, E.L.; supervision, E.L.; project administration, E.L.; funding acquisition, E.L. and E.A. All authors have read and agreed to the published version of the manuscript.

**Funding:** The reporting study by E. Lodygin was carried out within the scope of the research work of the Institute of Biology (No. 122040600023-8); E. Abakumov acknowledges Saint-Petersburg State University for a research project ID 93882802.

**Institutional Review Board Statement:** Not applicable.

**Informed Consent Statement:** Not applicable.

**Data Availability Statement:** Not applicable.

**Conflicts of Interest:** The authors declare no conflicts of interest. The funders had no role in the design of the study; in the collection, analyses or interpretation of the data; in the writing of the manuscript; or in the decision to publish the results.

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
