# Peer review of "The Use of Spectroscopic Methods to Study Organic Matter in Virgin and Arable Soils: A Scoping Review"

_agronomy, doi:10.3390/agronomy14051003_

Round 1

Reviewer 1 Report (Previous Reviewer 3)

Comments and Suggestions for Authors

In this paper the authors performed a systematic review on the use of NMR and EPR spectroscopy to study organic matter in soils. Compared to their previous submissions, here they define it as a PRISMA systematic review. However, even though they indicate now that they follow it, little has changed within the manuscript. To truly report your results, please see the checklist:

* For the abstract: http://www.prisma-statement.org/documents/PRISMA_2020_abstract_checklist.pdf

* For the study: http://prisma-statement.org/documents/PRISMA_2020_checklist.pdf?AspxAutoDetectCookieSupport=1

* For the flowchart: http://www.prisma-statement.org/PRISMAStatement/FlowDiagram

You have to specify all the above to make it easier for readers to follow your review.

For example, you have now defined some research questions. The organization of the manuscript has not changed substantially and I cannot immediately follow where each research question is answered and where the analysis results are. To address each research question, what kind of data did you extract from each paper?

As things stand, this review still is just an amalgamation of papers that present an overall picture of what has been done, much like an introductory chapter in a book.

Finally, in my humble view, since if you want to do a PRISMA systematic review much has to change, you could instead turn into a PRISMA scoping review: http://www.prisma-statement.org/Extensions/ScopingReviews which would allow you to perform fewer changes (but this still needs to be reported rigorously).

For the above reasons, and as I believe it is important to follow a structure in review papers to aid the readers, I regret to inform you that my suggestion is for a rejection & re-submission. I hope the authors understand my point and accept my suggestion.

Comments on the Quality of English Language

Mostly fine.

Author Response

Dear reviewer, thank you very much for your detailed work with the paper and for your suggestions and recommendations for improving the text of the manuscript.

We tried to translate our article into a PRISMA scoping review.
To do this, we characterised the article selection process in detail. Figure 1 "Flowchart of the search strategy and eligibility criteria" was added.
Section 3 NMR was divided into two subsections according to the questions posed in the articles. A detailed analysis of the articles according to the selected categories was also added to all sections.
All changes to the text are highlighted in yellow.

Yours sincerely,
Authors

Reviewer 2 Report (New Reviewer)

Comments and Suggestions for Authors

The text discusses the analysis of articles focused on the application of NMR and ESR spectroscopy in studying organic matter in soils. It highlights the widespread use of both solid- and liquid-phase NMR spectroscopy for investigating the structure of soil organic matter (SOM). While solution state NMR provides valuable insights into soluble SOM components, solid-state NMR offers advantages such as higher signal concentration and easier detection of non-protonized carbon atoms. However, solution NMR has limitations such as lower sensitivity and potential loss of structural information due to aggregation. EPR spectroscopy is praised for its high sensitivity in studying substances containing free radicals (FR) without altering their structure. The text emphasizes the importance of understanding FR structure in biopolymers to assess reactivity and transformations in various environments. It suggests that integrating two- and three-dimensional NMR spectroscopy with other physicochemical methods could provide a more comprehensive understanding of SOM composition compared to traditional one-dimensional methods.

I agree with the publication of this articol, after you answer two questions:

What significant advantages does EPR spectroscopy offer over other methods in experimental biochemistry?

In what way does the text suggest combining NMR spectroscopy with other physicochemical methods to enhance the understanding of soil organic matter composition?

Author Response

Dear reviewer, Thank you for reviewing our manuscript and for your comments.
In response to your comments, we have expanded the description of the advantages of EPR spectroscopy in the Conclusion. In the same section, we have added information on the advantages of combining NMR spectroscopy with chromatographic methods.
All changes in the text are highlighted in yellow.

Best regards,
Authors

Reviewer 3 Report (New Reviewer)

Comments and Suggestions for Authors

Ø   The current manuscript entitled "The Use of Spectroscopic Methods to Study Organic Matter in Virgin and Arable Soils: a Systematic Review" explores different techniques for analysing soil organic matter, with a particular emphasis on NMR and EPR spectroscopy.

Ø  The abstract is well-written and offers a thorough background introduction and study aim, but it lacks the decisive study findings.

Ø  The opening portion is eloquently crafted and very enlightening. The authors did not explain the aim behind starting this work.

Ø  Both NMR and EPR are well-established techniques with abundant data available regarding their advantages and disadvantages. What is the purpose of analysing the experience of utilising these two techniques when already several studies available?

Ø  Authors should conduct a meta-analysis while performing a systematic review. The current version appears dull. A basic analysis of the results is insufficient.

Author Response

Dear reviewer, thank you very much for your detailed work with the paper and for your suggestions and recommendations for improving the text of the manuscript.

The main findings of the study have been added to the abstract.
In the introduction, have been added the purpose of the study.
In all sections, have been added the analysis of the articles considered.

All changes to the text are highlighted in yellow.

Yours sincerely,
Authors

Reviewer 4 Report (New Reviewer)

Comments and Suggestions for Authors

General comments:

The article mainly analyzes the application experience of nuclear magnetic resonance (NMR) spectroscopy and electron paramagnetic resonance (EPR) spectroscopy methods in studying the molecular structure of SOM in mature and agricultural soils, as well as the advantages and disadvantages of different technologies in studying different SOM fractions (including humus preparations). This can provide a more comprehensive understanding of the molecular structure and properties of soil organic matter (SOM). However, there are some issues.

Specific comments:

1.     The abstract need to be rewritten. Add results to the abstract.

2.     Line 16 what is the meaning of mature soils? Do you mean the nature soils?

3.     The introduction needs more sufficient introduction according to the topic of this article, only describing the progress in researching chemical organic matter composition information, without describing the problems currently lacking in research. In addition, the purpose of this article is not clearly stated.

4.     Line 83 should be designed to address four issues.

5.     Legends 1 and 2 in figure 1 could be changed to NMR and ESR. The title needs more information. What is aim for the use of NMR and ESR? The figure could be made nicer.

6.     line 460-461, give the equation number.

Comments on the Quality of English Language

There are many very long sentences. It is difficult to understand and need to be improved. 

Author Response

Dear reviewer, thank you very much for your detailed work with the paper and for your suggestions and recommendations for improving the text of the manuscript.

1. The main results have been added to the abstract
2. Line 16 has been corrected.
3. Introduction has been corrected. The purpose of the paper has been added.
4. Line 83 has been corrected.
5. The designations 1 and 2 in the figure have been changed to NMR and EPR. Criteria for selection of articles and categories analysed have been added.
6. Equation number has been added in lines 460-461.

All changes to the text are highlighted in yellow.

Yours sincerely,
Authors

Round 2

Reviewer 1 Report (Previous Reviewer 3)

Comments and Suggestions for Authors

The authors have followed my suggestion to turn this into a PRISMA scoping review and have followed a good deal of the protocol. (Please understand that I want to help you so that it’s easier for next readers to understand your paper and accept my comments below as constructive.) However, some points still need to be addressed (I am using as a reference the PRISMA-ScR checklist https://www.prisma-statement.org/s/PRISMA-ScR-Fillable-Checklist_11Sept2019.pdf). As it stands, it’s not clear in each subsection 3.1 and 3.2 what each paragraph addresses and how these relate to the objectives stated. More structure is needed in my humble opinion.

1. For the abstract, PRISMA says: ‘Provide a structured summary that includes (as applicable): background, objectives, eligibility criteria, sources of evidence, charting methods, results, and conclusions that relate to the review questions and objectives.’ I am missing the eligibility criteria and the objectives (‘to analyze the experience’ is a bit too broad as you define 3 specific objectives).

2. The objectives should be in the introduction, not in the methods: https://knowledgetranslation.net/wp-content/uploads/2019/05/PRISMA-ScR_TipSheet_Item4.pdf

3. Line 100: you say four questions but only state 3. The third one needs re-phrasing as it is not in proper English.

4. Figure 2 has not proper placement w.r.t. its in text citation

5. Results of individual sources of evidences as charted in tabular format are missing: https://knowledgetranslation.net/wp-content/uploads/2019/05/PRISMA-ScR_TipSheet_Item17.pdf They should be given in an appendix

6. Limitations of the ScR are missing: https://knowledgetranslation.net/wp-content/uploads/2019/05/PRISMA-ScR_TipSheet_Item20.pdf

7. Lines 129 to 134: this is the qualitative information recorded from the articles, or the ‘data charting process’ and the ‘data items’. However, in the results, I don’t see a synthesis of them, e.g., in the form of boxplots, barplots or visual aids that can help identify for example what takes place. They are only sporadically given in the updated text you provided, but proper structure is missing. Perhaps you can consider re-arranging parts of the text. It’s also not clear how some of these results you present follow from the research questions posed by the ScR: for example, about the fundamental absorption peaks of the NMR spectra given in Figure 3. Or in Table 1, you could identify which or how many studies use each specific chemical shift. Perhaps you can give more structure to the text using additional subsections? Section 3.1 covers for example at least 4 pages!

8.        Case in point, the final two paragraphs of section 3.2 seem out of place and are difficult to point to which objective of your ScR they correspond to. This should be made abundantly clear and both sections 3.1 and 3.2 need to be organized better.

9.        The em dashes in Line 245 seem unnecessary?

10.  Line 148: recordsanalyzed (break into two words)

11.  Section 5 => Section 4

Comments on the Quality of English Language

Minor editing required as noted above

Author Response

Dear reviewer, thank you very much for your detailed work with the paper and for your suggestions and recommendations for improving the text of the manuscript.
1. The purpose and the criteria for the selection of articles have been added to the abstract.
2. The purpose has been moved to the introduction.
3. Corrected "four" to "three". The third question has been corrected.
4. Figure 2 is placed correctly. The link to it is before the figure.
5. Tables 1S and 2S showing the results of individual sources of evidence has been added as an appendix
6. Limitations of the ScR have been added
7. Additional figures have been added to the manuscript to facilitate analysis of the included articles. Table 1 has been slightly modified. Analyses of articles using chemical shift ranges have been added after Table 1. This information is also described in Table 1S in the Appendix.
Section 3.1. has been divided into five subsections.
8. Sections 3.1 and 3.2 reorganised. Some paragraphs deleted.
9. Dashes deleted.
10. Corrected.

Best regards,
Authors

Reviewer 2 Report (New Reviewer)

Comments and Suggestions for Authors

Thank you for the changes made, I agree with this form of the manuscript.

Author Response

Dear Reviewer,
Thank you for your detailed work on the article. Your suggestions and recommendations have improved our manuscript.

Best regards,
Authors

Reviewer 3 Report (New Reviewer)

Comments and Suggestions for Authors

I am satisfied with the author's response and endorse the immediate publication of this manuscript. 

Author Response

Dear Reviewer,
Thank you for your detailed work on the article. Your suggestions and recommendations have improved our manuscript.

Best regards,
Authors

Reviewer 4 Report (New Reviewer)

Comments and Suggestions for Authors

The authors have answered all questions carefully. 

Author Response

Dear Reviewer,
Thank you for your detailed work on the article. Your suggestions and recommendations have improved our manuscript.

Best regards,
Authors

This manuscript is a resubmission of an earlier submission. The following is a list of the peer review reports and author responses from that submission.

Round 1

Reviewer 1 Report

Comments and Suggestions for Authors

The content of this paper can represent a very useful summary of advantages and weak points of using NMR and EPR spectroscopy  techniques for studying different fractions of SOM. However, it contains the information of a review but it is not settle as a review.

In my opinion , a different structure of the paper should be presented.

A section of materials and methods should be added, including how many papers were investigated and  how papers were selected for being included in this paper (which queries were adopted, which databases were used, …). Into material and methods I think it could be useful for the reader to introduce the object of the review describing the principles on which the two techniques are based. Also the historical information that are now at the beginning of the paragraphs 2 and 3, can be removed from there, in order to make the current paragraphs 2 and 3 coincide with the section on advantages and weak points (that may be considered as results and discussions )

 I also would like to suggest to add a table for summarising the integral indices reported ( lines 241-258)

Reviewer 2 Report

Comments and Suggestions for Authors

The manuscript presents an interesting comparison between NMR and EPR techniques. It is well written, and the methods are thoroughly described. However, the authors should further develop the application aspect related to the evaluation of SOM using these methods.

To enhance clarity and organization, the authors should consider reorganizing the manuscript to include the following sections for each method:

·         Methodology:

o   Detailed description of the methodology used to explore different papers and manuscripts used in the study.

·         For each method (NM/EPR), the authors need to present:

§  Limitations and Potential: Discussion on the strengths and weaknesses of the method.

§  Field of Application: Specific applications of the method, such as SOM monitoring, SOM amendment evaluation, and soil health assessment.

§  Suitability for Different Soil Types: Discussion on the types of soils for which the method is most suitable, and factors influencing the method's efficacy in different soil environments.

Comments on the Quality of English Language

Moderate language correction is needed.

Reviewer 3 Report

Comments and Suggestions for Authors

In this review paper, the authors review the use of spectroscopy methods (NMR and EPR) to study organic matter in virgin and arable soils. To be honest, this whole paper would substantially improve if the authors followed the PRIMSA guidelines for systematic reviews. As it is, the paper reads more like a part of a PhD study, or an excerpt / chapter from a book that studies the past and working principles of NMR and EPR for organic matter detection but does not systematically review the bibliography.

While PRISMA is not a strict requirement in this journal, adhering to its principles can significantly improve the quality and transparency of systematic reviews. For example, the authors did not provide a clear rationale for their review process or the scope of their study, which could have been clarified by following PRISMA guidelines. It’s not clear if there are other reviews of these methodologies, nor why IR spectroscopy is not reviewed. It’s not clear which research questions this review paper attempts to answer.

The PRISMA guidelines offer several advantages:

Transparency: PRISMA guidelines improve the transparency of systematic reviews by providing a clear and structured framework for reporting methods and results. This transparency enhances the reproducibility of the review process and allows readers to assess the reliability of the findings. 

Quality: Adhering to PRISMA guidelines helps ensure the quality of systematic reviews by promoting a comprehensive and rigorous approach to study selection, data extraction, and synthesis. This can lead to more reliable and robust conclusions.

Standardization: PRISMA guidelines provide a standardized format for reporting systematic reviews, which makes it easier for researchers to compare and interpret studies. This standardization also facilitates the identification of gaps in the literature and areas for future research. 

Clarity: By requiring authors to clearly define the research question, inclusion criteria, and methods, PRISMA guidelines improve the clarity of systematic reviews. This clarity makes it easier for readers to understand the study design and findings.

Publication Bias: PRISMA guidelines include recommendations for assessing and reporting publication bias, which helps mitigate the impact of selective reporting and publication bias on the results of systematic reviews.

For the above reason, my recommendation to the editor is to reject the paper in its current form and urge you to resubmit it after following the PRISMA guidelines for systematic or scoping reviews.

Comments on the Quality of English Language

No comments.